# Artificial Intelligence vs. Natural Stupidity: Evaluating AI Readiness for the Vietnamese Medical Information System

**DOI:** 10.3390/jcm8020168

**Published:** 2019-02-01

**Authors:** Quan-Hoang Vuong, Manh-Tung Ho, Thu-Trang Vuong, Viet-Phuong La, Manh-Toan Ho, Kien-Cuong P. Nghiem, Bach Xuan Tran, Hai-Ha Giang, Thu-Vu Giang, Carl Latkin, Hong-Kong T. Nguyen, Cyrus S.H. Ho, Roger C.M. Ho

**Affiliations:** 1Center for Interdisciplinary Social Research, Phenikaa University, Yen Nghia, Ha Dong district, Hanoi 100803, Vietnam; phuong.laviet@phenikaa-uni.edu.vn (V.-P.L.); toan.homanh@phenikaa-uni.edu.vn (M.-T.H.); 2Faculty of Economics and Finance, Phenikaa University, Yen Nghia, Ha Dong district, Hanoi 100803, Vietnam; 3Sciences Po Paris, Campus de Dijon, 21000 Dijon, France; thutrang.vuong@sciencespo.fr; 4Vietnam-Germany Hospital, Hoan Kiem district, Hanoi 100000, Vietnam; kimcuongvd@gmail.com; 5Institute for Preventive Medicine and Public Health, Hanoi Medical University, Hanoi 100000, Vietnam; bach.ipmph@gmail.com; 6Bloomberg School of Public Health, Johns Hopkins University, Baltimore, MD 21205, USA; carl_latkin@jhu.edu; 7A.I. for Social Data Lab (AISDL), Vuong & Associates, Dong Da district, Hanoi 100000, Vietnam; htn2107@caa.columbia.edu; 8Institute for Global Health Innovations, Duy Tan University, Da Nang 100000, Vietnam; giang.ha.18@gmail.com; 9Center of Excellence in Artificial Intelligence in Medicine, Nguyen Tat Thanh University, Ho Chi Minh City 100000, Vietnam; gigi.vugiang@gmail.com; 10Department of Psychological Medicine, Yong Loo Lin School of Medicine, National University of Singapore, Singapore 119228, Singapore; pcmrhcm@nus.edu.sg; 11Department of Psychological Medicine, National University Health System, Singapore 119228, Singapore; su_hui_ho@nuhs.edu.sg

**Keywords:** artificial intelligence, AI in healthcare, AI in medicine, AI readiness, Vietnam, AI applications

## Abstract

This review paper presents a framework to evaluate the artificial intelligence (AI) readiness for the healthcare sector in developing countries: a combination of adequate technical or technological expertise, financial sustainability, and socio-political commitment embedded in a healthy psycho-cultural context could bring about the smooth transitioning toward an AI-powered healthcare sector. Taking the Vietnamese healthcare sector as a case study, this paper attempts to clarify the negative and positive influencers. With only about 1500 publications about AI from 1998 to 2017 according to the latest Elsevier AI report, Vietnamese physicians are still capable of applying the state-of-the-art AI techniques in their research. However, a deeper look at the funding sources suggests a lack of socio-political commitment, hence the financial sustainability, to advance the field. The AI readiness in Vietnam’s healthcare also suffers from the unprepared information infrastructure—using text mining for the official annual reports from 2012 to 2016 of the Ministry of Health, the paper found that the frequency of the word “database” actually decreases from 2012 to 2016, and the word has a high probability to accompany words such as “lacking”, “standardizing”, “inefficient”, and “inaccurate.” Finally, manifestations of psycho-cultural elements such as the public’s mistaken views on AI or the non-transparent, inflexible and redundant of Vietnamese organizational structures can impede the transition to an AI-powered healthcare sector.

## 1. Introduction

“My colleagues, they study artificial intelligence; me, I study natural stupidity.”—Amos Tversky (1937–1996)

Artificial intelligence (AI), including its potential impacts and threats, has stopped to be a far-fetched topic that only scholars and visionaries engage with. The surprising achievements of the field, such as the sensational victory of AlphaGo against the best human Go player, have indeed captured the imagination of the wider public. In the broadest sense, AI is an inter-disciplinary approach using principles and devices from computation, mathematics, logic, mechanics, and even biology to solve the problem of understanding, modeling, and replicating the intelligence and cognitive processes [1]. Driven by the big data revolution and the falling cost of computing power in recent years, AI has suddenly become much better at all kind of tasks and become more general in their applications [2,3]. One of the most practical applications of AI is in the field of medicine, fueling what could be called ‘computational medicine’ [4]. A major driver of the continual interest in AI is the successful application of deep learning, which refers to the ability of an artificial neural network with many layers to train themselves on recognizing patterns and classifying huge datasets [5]. As such, the use of predictive and cognitive analytics or deep learning in medicine is expected to improve accuracy in designing personalized treatments [6] or predicting heart attacks [7]. In addition, the symbiosis of AI-technologies and human clinicians has been noted in by researchers in the field [8]. The United Nations (UN) notes that when human diagnosis is accompanied with AI, the error rate is lowered to 0.5%, versus 3.5% for human doctors [7], recent studies have also shown the synergy of human clinicians and AI yield better results than either alone [9]. In fact, applications of AI are already transforming the healthcare sector both in clinical and community settings thanks to their promising results, such as in health information systems, geocoding health data, epidemic and syndromic surveillance, predictive modeling and decision support, and medical imaging [5,8,10,11,12].

Given the new technology’s power to save more lives, it is no surprise that the UN itself has begun discussing how to advance the sustainable development goals (SDGs) worldwide with the applications of AI in “primary care, service delivery, integration and analysis of medical data, and responses to disease outbreaks and other medical emergencies,” in line with the third SDG on good health and well-being [13]. Yet, at the same time, as this paper will show, research and major adoptions of health intelligence have mostly taken place in developed countries where the discourse itself began very early on and the technological and medical infrastructure has matured sufficiently. What is left for discussion is the preparations needed to apply the advanced technologies in medicine in less developed countries. To achieve the SDGs set by the UN under the new context, policymakers, especially those from developing countries, should be aware of the tasks ahead, including the level of a country’s AI readiness as well as its challenges and barriers. It is notable that, despite the voluminous literature on AI in general and AIM in particular, there seems to be little or no research yet on the evaluation of a country’s readiness to implement such technologies. Moreover, although there is abundant research on AI applications in high-income countries, such studies in resource-poor settings are more limited [14]. To contribute to the literature of AIM in the developing world, this paper reviews the use of AI in medicine across the globe, and then examines the status of AI readiness in Vietnam, a populous developing country in Southeast Asia with a per capita income of approximately $2000 and intensifying demand for better healthcare [15,16].

Through the case study of Vietnam, this study also looks into the correlation between economic development and AI research and applications. According to various reports, the biggest and most important AI ecosystems are located in the U.S., China, Germany, France, and the Nordic region—all of which are countries with rich resources for AI research and development [17]. In terms of market share, in 2017, North America accounted for the largest market share and is expected to remain the leader through 2025 due to the availability of capital investments, government funding as well as the presence of leading players and strong technical base [18]. Estimates from the International Data Corporation (IDC) confirm that, geographically, the U.S. is expected to account for 60% of all spending on cognitive/AI systems by 2022, followed by Western Europe and China [19]. Indeed, there seems to be a direct correlation between the growth of the AI market worldwide, including in medicine and healthcare, and the economic conditions, such that a higher level of disposable income could propel the spending trends in the field [20]. This is indeed a difficulty facing developing countries, which are already in need of high technologies to control the widespread of communicable diseases but lack the resources to compete with the developed counterparts.

The structure of the review paper is as follows: the literature review section will provide an overview of the research and applications of AI in medicine (AIM) throughout the world; then the framework section will present the conceptualization used to evaluate the AI readiness; the next section will delve into the case of Vietnam, touching on both the overall AI readiness and the particular readiness of the healthcare sector. The final section will discuss the findings and tie up any loose ends, with suggestions for future research direction.

## 2. Literature Review

### 2.1. Advances in AIM Since 1950s

The British mathematician Alan Turing is one of the founders of modern computer science and AI, having proposed in 1950 that any artificial intelligent system passing the Turing test is one that has the capacity to generate human scale performance [21]. In the field of medicine, the goal of computer-aided programs is essentially to simulate expert human reasoning, thereby, improving the clinical diagnosis and treatment process. The IBM Watson system, a pioneer cognitive program in this field since 2010, is a notable example of digital computing strengths for its personalized assistance in oncology diagnosis and treatment [22,23]. Given the potential advantages of such systems, applications of intelligent techniques in medicine have flourished since the middle of the last century [24]. This section will review some of the major breakthroughs on artificial intelligence in medicine (AIM).

Research on computer-aided diagnosis began in the United States the 1960s, with initial attempts at applying Boolean algebra, pattern matching, and decision analysis to the diagnostic process—all of which yielded little practical value due to its narrow clinical focus [25]. Such work was then discarded, making way for a new line of research in the 1970s that focused more on modeling human clinical expertise, either through the rule-based or the matching strategies systems [25,26]. For instance, in 1976 Gunn successfully used computer analysis to diagnose acute abdominal pain for patients needing surgery [24]. Yet, the majority of efforts to emulate models of clinical problem solving were met with disappointment, especially in difficult medical cases, because of the rule-based programs’ costliness and inherent lack of pathophysiological knowledge [5,25]. Over the next decade, interests in developing knowledge-based systems in medicine grew. By the 1990s, the AIM field had matured significantly thanks to numerous research and development activities, which helped fuel the growth of biomedical informatics [26,27].

While classification by AI varies considerably, AIM can be examined through two main branches: (i) the virtual branch which includes informatics approaches, and (ii) the physical branch which involves assistive robots in elderly care, surgery, etc., as well as nanorobots in drug delivery [11]. The virtual one is represented by machine learning and mathematical algorithms, which are divided into three types: unsupervised—pattern recognition, supervised—classification and prediction, and reinforcement learning—strategic sequencing of rewards and punishments [11] (p. 6). Within the virtual branch, the use of AI devices in the health sector can be categorized as: (i) devices using machine learning techniques that analyze structured data such as imaging, genetic, and electrophysiological data, and (ii) devices employing natural language processing methods that get information from unstructured data such as clinical notes [23]. The two methods are complementary given that the latter procedures can turn texts into machine-readable structured data, which can be used by machine learning techniques [23].

Ramesh, et al. [24] provide a useful overview of the four major AI techniques that are most often applied in the virtual branch and resulted in important clinical applications over the years. Table 1 recaps this information.

As can be seen, a significant portion of research in AIM has pointed out the useful application of AI techniques in the virtual branch, such as in medical records analysis, expert consultation systems, clinical diagnosis and related decision support tools, to name a few. Out of all the techniques, applications of artificial neural networks (ANNs) appear to be the most prominent in medical science thanks to their capacity to simulate the neurological processing ability of the human brain [24,28], as well as to learn, tolerate data noises, and model incomplete data [29]. The wide-spread use of ANNs is thanks to the advancement in computational power, which allows modern neural networks’ deep learning to run through hundreds of millions of parameters [5]. In clinical settings, the application of this network is most visible in the adoption of decision support systems from the 1970s in the U.S., or the electronic health records system that focuses more on clinical data [5,24]. At the same time, given the high level of uncertainty in medical diagnosis, the application of fuzzy techniques also provides a powerful decision support tool, such as in the early diagnosis of pathologies [29].

Within the virtual branch of AIM, specifically in bioinformatics, increasing attention is paid to the role of data-centric approaches, known as big data analytical tools and applications [30,31,32,33,34,35,36]. This focus is partly driven by the integration of genomic data into patient care, opening up the opportunities for clinical phenotyping patterns as well as posing serious challenges in the management of big data [31,34]. According to Chute, et al. [31], this development is “attributable to advances in chip-based genotyping and next-generation sequencing (NGS) methods.” The complex, large-scale data generated during clinical care, in addition to the electronic health records, can be useful in translational bioinformatics [35], cardiovascular disease research [30,34], deep patient genotype and phenotype identification [33], clinical pharmacology [36], or even in smartphone-based digital phenotyping to improve mental health [32]. In population-scale human phenotyping, the cautions remain that results in well-designed experiments are not always readily replicated and that there is increased burden of proof in mining data with layers of multiplicity [37].

In terms of developments in the physical branch, while many biomedical laboratories have used AI-controlled robotic systems in assembly line, the adoption of autonomous robots in medical interventions, often known as robotically assisted surgery, is taking place at a slower pace [5,11]. Some examples of robotic surgery, which still requires a surgeon for movement control, include the not-autonomous U.S. government-approved da Vinci surgical system for minimally invasive procedures, or a supervised autonomous robotic system for suturing an intestinal anastomosis, or other autonomous robots for cochleostomy [5]. On-going research on the pre-programmed, image-guided and teleoperated surgical robots hold out the promising prospect of more automated medical interventions in the near future. Overall, by combining the technologies with the physicians, diagnostic confidence as well as surgical procedures would improve reliably, resulting in fewer medical errors, more optimized care trajectory for chronic disease patients, precision therapies for complex illnesses, and improve subject enrollment in clinical trials [7,38].

### 2.2. Research and Applications of AIM in Non-Western Countries

Studies on the development and breakthroughs in AIM since the 1950s have largely been centered in the U.S. and similarly developed countries. This observation reflects the larger trends in AI research worldwide. In terms of academic publications, according to Elsevier’s survey of global research output on AI from 1998 to 2017, Europe as a whole remains the biggest contributor to the field but is losing ground as the U.S. and China gear up to overtake this role [39]. Using data provided openly by Elsevier [39], this paper reframes the publications on AI in relations to the country’s GDP per capita (World Bank statistics), as captured in Figure 1.

Two observations stand out in Figure 1. First, China and India are the sole two countries with GDP per capita in the upper middle and lower middle-income levels, respectively, to be ranked in the top ten countries worldwide with the greatest number of AI publications. Second, economic wealth has not necessarily led to increased research on AI in the past decades, given that many other developed European nations are not listed in the top 20 of this list [39]. What also matter are the political drive and commitment to develop the technological infrastructure, as what would likely be the cases of several developed countries with the number of publications hovering between 20,000 and 40,000 during the period.

Against this background on the overall AI research landscape, this section reviews the research on and the use of AI technologies in medicine in non-Western countries, including both the high- and low-income ones. Here, a notable trend stands out: while AIM in high-income countries takes places mostly in clinical or laboratory settings, with the focus on certain types of diseases such as cancer, nervous system disease, and cardiovascular disease [23], the emphasis is quite different in the other parts of the world. Studies from upper middle- to high-income countries in Asia such as China, South Korea, and Japan show a strong rise in the applications of AI technologies in the healthcare sector, closely following the trends of AIM breakthroughs in the developed world. By contrast, the applications of AIM in low- and middle-income countries will be shown below to be more public health-oriented rather than clinically-focused.

In particular, the adoptions of digital technology aimed at improving health behavior or assessing health performance were seen in the analytical databases for dietary fiber in South Korea [40], the intelligent performance assessment system for nursing staff in elderly home care services in Hong Kong [41]. To predict influenza epidemics, a study constructed a model using support vector machine for regression to analyze search queries on South Korean social media [42]. In more advanced moves, a Chinese team used deep-learning algorithms to create an AI agent capable of accurately and efficiently supporting the screening of congenital cataracts as well as the treatment decisions by ophthalmologists across multiple hospitals [4]. Other AIM research projects coming from China have touched on the feasibility of using support vector machine model combined with metabonomics to check the safety of traditional Chinese medicine [43], or applying computational methods such as data mining to modernize traditional Chinese pharmacy [44,45,46,47]. By comparison, AIM studies in Japan show an advanced level in the physical branch, such as the successful robot assistance in thoracoscopic surgery [48], elderly care [49,50], or the health exercise demonstration robot TAIZO that can encourage more people to work out [51].

At a national scale, the Chinese government is implementing a 15-year plan titled “Brain Science and Brain-Inspired Intelligence” (known in short as the China Brain Project), which does research on the neural circuit mechanisms underlying cognition as well as the interactions between brain disease diagnosis/treatment and brain-inspired intelligence technology [52]. In a larger scheme, China’s National Development and Research Commission has also launched a national engineering laboratory for deep learning, led by Baidu [53]. These plans are in line with China’s ambition to become the leading investor in cognitive/AI systems; its compound annual growth rate (CAGR) for the 2017–2022 forecast period will be 43.8%, somewhat behind Japan’s 62.4% CAGR [19]. Meanwhile, the state-run Korean Bio-Information Center plans to put into operation the National DNA Management System which, by integrating massive DNA and personal medical records, will provide customized diagnosis and medical treatment [54].

Contrary to the clinical focus of AIM applications in high-resource countries, it appears that AIM in low- and middle-income countries is often related to automation technology, such as the mobile phone framework, concerned with the delivery of health information and health communication that could improve public health outcomes [16,55,56,57,58]. For example, in Tanzania, the electronic version of the Integrated Management of Childhood Illness (IMCI) protocol, dubbed e-IMCI, has allowed local clinicians to use personal digital assistants (PDAs) to classify and treat child illnesses, thereby in the long run, contributing to better pediatric healthcare [59]. This expert system, which integrates the second generation protocol specification format GuideLine Interchange Format (GLIF), is particularly suited to low-income settings given that computer systems training for IMCI were 23–29% less expensive and as effective as standard training methods [59]. Not only would PDA and mobile phone-based tools improve the scope and efficiency of field health workers in low-income regions, they would also be of tremendous help in monitoring and containing chronic diseases and communicable diseases thanks to their wide availability [57,58]. Under these programs, also known as ‘e-health’, governments in several low- and middle-income countries have successfully implemented community information systems for disease surveillance [56]. Although more research is needed on the impacts of e-health on outcomes and costs in these countries, several steps, such as text messaging for improving patients’ self-care and automated telephone monitoring, have been shown to have positive outcomes in chronic disease management [56]. Besides mobile health technologies, case studies from clinics in Honduras and Mexico suggest that a cloud computing model using automated self-management calls plus home blood pressure monitoring could improve outcomes for hypertensive patients in low- and middle-income settings [60]. This means even areas with limited infrastructure for patient-focused informatics support could apply this model at their telecommunication centers [60]. In an attempt to better monitor risks of public health emergency, a study has suggested low-resource countries also adopt ‘syndromic surveillance’, which uses pre-diagnostic data and statistical algorithms to rapidly detect epidemics and characterize unusual morbidity trends [61]. Some examples of this approach could be seen in the Early Warning Outbreak Recognition Systems (EWORS) that have been in place in Jakarta, Indonesia since 1998 and in Lima, Peru since 2005—both surveillance systems proven to be effective in detecting early disease outbreaks [61].

This section by no means offers an exhaustive list of the on-going use of AIM in the non-Western parts of the world. It does, however, point out an interesting trend regarding the direction of AIM development and applications through the lens of a country’s resource level (financing, human, and infrastructure). For instance, the biggest and most important AI ecosystems are located in the U.S., China, Germany, France, and the Nordic region—all of which are countries with rich resources for AI research and development [17]. This observation forms the premise of this paper’s inquiry, which is on the AI readiness criteria. The next section will delve further into the framework of this question.

## 3. Framework

To understand the criteria required for a successful implementation of AI technologies in the healthcare sector, one needs to be familiar with the challenges in the field of AIM and the components of a functional AI system.

In the discussion about the challenges of applying AI in medicine in the 1990s, Shortliffe [27] has outlined three key points, namely that (i) AIM cannot be isolated from the rest of biomedical informatics nor from health planning and policy, (ii) the applicability of AIM in real-world settings depends on the integration of various knowledge-based tools, rather than the stand-alone consultation systems of the 1980s, and (iii) politicians and policymakers need to understand the strategic role of and, thus, investment into clinical and biological computing infrastructure. In addition, other scholars have urged health information technologies to recognize the cognitive factors that determine how human beings comprehend information, solve problems, and make decisions [26]. Particularly, Patel, et al. [26] suggested AIM systems be resilient enough to address the human errors and risk taking.

Many of these issues are being addressed as the first generation of knowledge-based AI systems gave way to more advanced systems that can account for more complex interactions and pattern recognition [5]. What remains of significant concern is the investment in computational medicine, such as to build data centers that can process the growing volume of data generated from clinical settings. For instance, in China, attempts to modernize and digitize the field of traditional Chinese medicine (TCM) call for the construction of a large-scale clinical data warehouse system, one that integrates the structured electronic health records for medical knowledge discovery and TCM clinical decision support [62,63]. For low- and middle-income countries, including Vietnam, even when policymakers understand that operating a coherent national informational system will increase efficiency and decision-making, thus, leading to better health outcomes, there lie complex challenges in building such a system of the national scale. On the architectural frameworks for developing national health information systems in low-resource countries, a study outlines four key requirements, namely (i) coordinated approach to deal with the organizational complexity, (ii) collaborative approach to deal with autonomy and heterogeneity, (iii) an evolutionary approach to deal with dynamism of the models, and (iv) interoperability over integration [12]. Besides the organizational aspects, a government also needs to take into account the technical, financial, political, to ethical, societal, and cultural requirements for building any national informatics system. For instance, to set up an electronic syndromic surveillance system, Chretien, et al. [61] have noted an extensive list of considerations that include but are not limited to: initial and refresher system training, multilateral technical partnerships, continual funding, local political support, participation of key political stakeholders, community education, etc. These issues are not new to any developing countries but they are indeed critical to the successful implementation of any high technology programs in medicine.

Based on the literature, this paper sums up the four considerations for successful AIM applications in Figure 2. The model calls for a combination of technical or technological expertise, financial sustainability, and socio-political commitment, all are embedded in a healthy psycho-cultural context. In technical terms, as machine learning methods, especially the artificial neural networks, rely on the availability of large volumes of high-quality data, a strong and stable technological infrastructure, such as large data centers or warehouses, is needed [5]. To maintain such a system requires continual financial resources and socio-political support. More specifically, the priorities should include investing in AI research and development, building an AI ecosystem, and encouraging cross-industry partnerships that would make AI applications both more accessible and less costly to adopt. The psycho-cultural context can either act as the accelerator or decelerator of the transitioning to AI-powered healthcare sector.

As shall be seen in the case of Vietnam in this paper, when one or two of these components are missing, any talks of AIM would remain theoretical. The AI readiness of a country, therefore, is assessed by its preparations and investment in the technological infrastructure, followed by its political commitment to better both clinical medicine and public health.

## 4. Evaluating AI Readiness in Vietnam

### 4.1. Evaluating the Overall AI Readiness

The first criterion to assess the level of AI readiness is the quantity and quality of AI-related research in a country. For Vietnam’s academic research on AI or using AI techniques, Elsevier [39] reports that Vietnamese researchers had published 1556 papers during the 1998–2017 period. Figure 1 has shown the absolute number of papers from Vietnam is small compared with that in countries of similar GDP per capita such as India, Brazil or Egypt. The data unfortunately do not give any chronological or subject breakdown of the publications.

However, a closer look at the literature shows that AI-related research in Vietnam has been done in many different fields: geosciences [64,65,66,67,68,69,70,71], speech recognition and machine translation [72,73], wireless broadcasting [74], civil engineering [75], and medicine such as the predictions of diseases-correlated genes [76,77] and communicable diseases monitoring and control [78,79,80,81] (see Appendix A: “Review of Vietnamese Publications on AI.xlsx” for more details).

In terms of methodology, the specific techniques vary widely across the fields, though support vector machines (SVM), machine learning, or hybrid approach appear more prevalent. For example, geoscientists in Vietnam have tried to map landslide susceptibility using adaptive neuro-fuzzy inference system [65] or SVM, decision tree, and Naïve Bayes (NB) models [64,66], or to model flood susceptibility and forest fire susceptibility using hybrid AI approach [67,68]. They have also applied a new hybrid intelligent method based on least squares support vector machines (LSSVM) and artificial bee colony (ABC) optimization, both of which are “state-of-the-art soft computing techniques” rarely used in landslide susceptibility assessment [70]. The hybrid AI approach based on metaheuristic and machine learning was also utilized in a study to assess slope stability in civil engineering [75]. Others have used SVM and ANN to classify images [82], to classify trees or model the presence of macroinvertebrates in rivers [83], or the Bees algorithm to improve university timetabling [84], simulation-gaming and multi-agent modeling to analyze Vietnamese land use systems [85,86], satellite-derived and Geographic Information Systems (GIS)-based information to map solar resources and power potential in Vietnam [87]. In the studies about speech recognition and machine translation, researchers have applied a method based on deep learning to extract bottleneck features for large Vietnamese vocabulary speech recognition [72].

It appears that in Vietnam, though the number of sheer publications in AI might be small, the Vietnamese experts could employ the state-of-the-art AI techniques in a variety of fields. With the emphasis and embrace of the government on the concept of Industrial Revolution 4.0 and of high-tech start-ups or computational entrepreneurship [88,89], of which AI is a core component, it is likely that basic research in AI as well as its applications will thrive there. At the first ever conference on Artificial Intelligence in Vietnam, AI4LIFE 2018, Dr. Nguyen Huu Duc, the vice-rector of Vietnam National University in Hanoi, remarked that there are currently 140 training programs on information technology in Vietnam, of which 57 courses are related to AI. Though the number is small, this first conference, which brought together the Ministry of Science and Technology, the Vietnam Union of Science and Technology Associations, many scholars from developed countries as well as the AI industry, is expected to create a shared orientation for developing and applying AI techniques in the country [90]. All signs seem to point to a great potential for the use of AI in Vietnam. The next section will attempt to evaluate specifically the AI readiness in the healthcare sector. 

### 4.2. Evaluating the AI Readiness in Vietnam’s Healthcare Sector

#### 4.2.1. Research on AI in Medicine and Health in Vietnam

The previous section has pointed out that Vietnamese physicians do conduct research on AI in medicine and publish their studies in international journals. It also appears that many Vietnamese researchers in healthcare and medicine are capable of applying the state-of-the-art AI techniques classified by Yu, et al. [5]. Of the techniques utilized, the convolutional neural network has been used to classify human organs images and automatic extraction of chemical-induced disease relation [91,92], the unary classification, binary semi-supervised and positive and unlabeled learning classifications were used to help predict diseases-associated genes [76,77], remotely-sensed data, GIS, spatial analysis, and machine learning classifiers show the potential for malaria vulnerability mapping and spatial disease control [78,79,80]. Other researchers have developed predictive forecasting algorithms to track the spatiotemporal trends of diarrheal disease hospitalization in Ho Chi Minh City [93]. On ways to improve the health sector’s informatic system, there are suggestions to build a centralized database with decentralized management to recode, analyze and monitor the data of anti-retroviral (ARV) therapy and HIV patients across different Vietnamese hospitals and healthcare units [94], or to optimize the K-means clustering operation in data mining [95]. In a more sweeping attempt, researchers have assessed the pharmacovigilance status in ten Southeast Asian countries, including Vietnam, using a quantitative signal detection algorithm [96]. Vietnam, in its economic success, is known for an extensive international cooperation and reformation for human resources development [97,98]; thus, it is no surprise that the healthcare sector, one of the most technological advanced sectors, does have advanced technical experts.

Though this human capital element factors in positively for the AI readiness of the healthcare sector in Vietnam, it appears that the government has not fully tapped this potential. This is evidenced by the fact that the majority of the studies being review were not funded by any state organizations, implying the lack of a national coordinated effort such as the case in China [39,52,53]. Among the works, only the study by Le and Nguyen [76] was made possible thanks to funding from the Vietnam National Foundation for Science and Technology Development (NAFOSTED), an arm of the Vietnamese Ministry of Science and Technology. Another research by Pham, et al. [80] received financial support from the Vietnam Academy of Science and Technology, a ministerial level research institute. Moreover, many of the publications are the results of international cooperation which suggests the efforts of individual Vietnamese researchers and their institutions to catch up with the worldwide trend in AI research. Successes from these isolated attempts could be greatly amplified if the socio-political will is aligned.

#### 4.2.2. Applications of AI-Related Technology in Vietnam’s Healthcare

To evaluate the readiness to implement AI technology in medicine and healthcare in Vietnam, this paper first conducted text mining of the annual reports published by the Vietnamese Ministry of Health from 2012 to 2016 [99,100,101,102,103]. All the reports are available online [99,100,101,102,103]. The main methodology is running tm package for text mining in R software (version 3.4.1) to look for the frequencies of five certain keywords, namely ‘data’, ‘database’, ‘technology’, ‘information’, and ‘technical’, that are related to AI or high technology (see the Appendix A: “R codes.docx”). The results, summarized in Figure 3, highlight the lack of information databases in the sector as well as any relevant technical or technological applications. For instance, in the 2012 Joint Annual Health Review (JAHR) [99], when the words ‘health’ (which appeared 2169 times), ‘medical’ (894 times), and ‘hospital(s)’ (837 times) were removed from the search, the list of top ten most used words in the report includes only one keyword, ‘information’ (266 times) being tracked (see the Appendix A: “MOH Reports 2012-2016 data.xlsx” for more details). Figure 3 presents the top ten most frequent words in this report while Figure 4 shows the frequencies of the five tracked keywords in the Ministry’s annual reports during the five-year period.

The results highlight the fact that information is abundant but the technological system to store, process or analyze such information is not yet formulated. The word ‘database’ was so scarcely mentioned in the report that it begs the question of how the data are being used in any useful manner to serve the public health. To further confirm the observation on the lack of database or data center, the study ran another line of code to find the probabilities of other words appearing with the word ‘database’ in the reports throughout the years, taking into account only words with correlation higher than 0.3. Table 2 shows the probabilities of some words that often appear with ‘database’.

Further in-depth reading of the reports revealed the extent to which a digital database system is absent in Vietnam’s healthcare sector. Particularly, while most hospitals have put into operation their own internal management systems, they acknowledged that these systems remain primarily focused on human resources and related logistic issues rather than clinical subjects. As such, there is still no system to monitor the diagnosis and treatment, prescription of drugs, or to report/raise warnings on medical errors, or to track the spread of communicable diseases at the national level, etc. The reports pointed out the lack of standardization in applying the International Statistical Classification of Diseases and Related Health Problems, a medical classification list by the World Health Organization. Many reasons can account for these shortcomings, such as: unstructured data collection at all levels nationwide, missing legal guidelines and instructions, investment in information technology in the sector being spread too thin and rendered ineffective, substandard IT manpower, and low-quality research and training on IT in medicine in Vietnam.

In the 2016 JAHR, the Ministry of Health said it has begun digitizing administrative tasks, including using digital signatures, and collaborated with other agencies to boost the applications of IT in medical treatment and health insurance payment. This is in line with the Ministry’s plan, which was issued in September 2016, to improve the IT system of the diagnosis-treatment-insurance payment process. There is no mention of an automated system in the sector yet. Overall, the big issue is the lack of an integrated and unified system that adopts international standards in managing both the clinical and administrative processes. Based on the reports, Table 3 recaps the areas in which Vietnam’s healthcare sector has sought to apply IT.

To account for the aforementioned lack of readiness to apply AI technologies in medicine and healthcare, one first needs to be familiar with the direction of information flows within the Vietnamese health sector. All information reported to the Ministry of Health came in a one-way direction from the lowest tier levels through a paper-based system, through districts and onto provincial or municipal health departments before it would be entered electronically for data compilation and management at the upper ministerial levels [104]. This system of information reporting and processing not only results in the loss of valuable information along the way (due to possibly incompetent staff at the lower tiers) but also in the inaccuracy of any final reports at the upper levels. As such, it would render any efforts to build a national health information database futile. Despite the ministry’s efforts to promote IT applications over the past decade, most medical units that use electronic health records or similar technologies at the provincial level have not reached the full usage potential of these e-health systems [104], instead wasting resources by manually taking on managerial activities. In a review of the challenges plaguing the Vietnamese healthcare information management system, Nguyen, et al. [104] pointed out that, although the system was first initiated back in 1995 and re-introduced with improvements in 2002, its usage nationwide remains incomprehensive, non-transparent, and spontaneous. One of the reasons might be the lack of funding—the 2002 version of the system was adopted in small scales thanks to several international donors such as the WHO, the Asian Development Bank (ADB), and the UN [104].

However, with an impressive economic performance over the past three decades [97,98] and the government’s embrace of the Industrial Revolution 4.0 concept [88], the financial and socio-political factors will become increasingly favorable. The uncertain factor that could determine the advance of AIM in Vietnam will be the psycho-cultural background. The government and even the whole society can be in favor of a wider and more effective use of AI techniques, yet, the organizational and attitudinal changes that are necessary for this shift might take a long time. First, similar to Industry 4.0, there have been many misconceptions and misuses of the AI concept in the Vietnamese media. Managing the expectations of the public will be crucial for the successful transition to an AI-powered healthcare sector in Vietnam, as it is also true all over the world [105]. Second, several studies have confirmed that the structures of Vietnamese state organizations, including the healthcare sector, are usually non-transparent, inflexible and redundant with many departments performing the same functions [104,106,107]. Meanwhile, AI must be applied successfully in the public sector to reach the highest utility since the heavy reliance on this sector is one of the outstanding features for healthcare delivery in Vietnam, according to the Japan International Cooperation Agency [108]. These undesired aspects in the psycho-cultural elements need to be addressed to improve the AI readiness in Vietnamese healthcare.

## 5. Discussion and Conclusions

Based on an extensive review of the literature on the topic, this paper has presented a framework to evaluate the AI readiness for healthcare sector in developing countries: a combination of adequate technical or technological expertise, financial sustainability, and socio-political commitment embedded in a healthy psycho-cultural context could bring about the smooth transiting toward an AI-powered healthcare sector. Using the Vietnamese healthcare sector as a case study, this study has reviewed the development and applications of AI techniques here. 

The study has found that the Vietnamese experts have employed some of the state-of-the-art AI techniques in their research, which indicate the readiness for AI in terms of human capital. However, there remains a lack of socio-political commitment, hence, the financial support to amplify the fruit of this human capital element. More specifically, the majority of the reviewed publications on Vietnam’s AIM seem to be not funded by any state organizations. Though the efforts of individual researchers and their institutions have kept this small community abreast of the trend in world, one can argue that a more coordinated support from the government can advance the field much further.

The more serious challenge to AI readiness of Vietnam’s healthcare sector is the weakness in building data centers and data management. Several efforts for creating standardized data warehouses for healthcare in Vietnam have failed to yield the desired results even though they were initiated in 1995 [104,107]. Using the text mining of the official annual reports from 2012 to 2016 of the Ministry of Health (Table 1), the paper found that the frequency of the word ‘database’ actually decreases from 2012 to 2016, and the word has a high probability to accompany words such as “lacking”, “standardizing”, “inefficient”, and “inaccurate” (Table 2). A further in-depth analysis of the use, or lack thereof, of information technology in Vietnam’s healthcare sector reveals most databases, whether for administrative, clinical, or national purpose, are still either under construction or not yet built. In the case of electronic medical records (EMRs), one of the most valuable resources for the application and development of machine learning and other AI tools, most Vietnamese facilities have used EMRs to store and manage patients’ data, yet, the data output does not share the same digital language [99,100,101,102,103]. This challenge of data standardization is not unique to Vietnam as Faust, et al. [6] and He, et al. [9] have shown that this issue is among the critical barriers for the real-world clinical application of AI-technologies.

The study also argues that underlying all the technical challenges, there are also the psycho-cultural barriers. The psycho-cultural context can influence the perspectives and behaviors of a society, especially in dealing the highly uncertain terrain such as AI technologies [109,110,111]. On the one hand, as investment into AI research or infrastructure for AI can be expensive, the irrational perspective of the public can slow down substantially the transitioning to an AI-powered healthcare sector [112], the expert community and science communication circle arguably must not ignore this problem of expectation management [105], as Topol [8] pointed out “The state of AI hype has far exceeded the state of AI science.” On the other hand, any improvements in AI readiness in Vietnam would require addressing the problem of non-transparent, inflexible and redundant organizational structures of Vietnamese state organizations, including in the healthcare sector [104,107].

As remarked in a keynote speech in the Industry 4.0 Summit on July 13, 2018 in Hanoi, “Government will not be a partner to Industry 4.0 if it is stuck in Bureaucracy 1.0” [113]. It is an irony that the threat of getting stuck in Bureaucracy 1.0, an example of natural stupidity, is perhaps that biggest threat to the advancement and success of artificial intelligence in developing countries such as Vietnam. This observation of the keynote speaker in the Summit echoes the wisdom of Amos Tversky: “My colleagues, they study artificial intelligence; me, I study natural stupidity” [114].

## Figures and Tables

**Figure 1 jcm-08-00168-f001:**
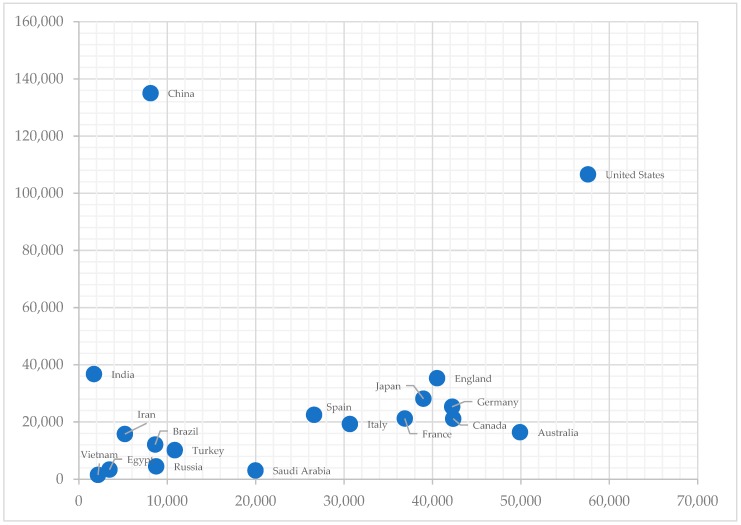
A visualization of the relationship between Gross Domestic Product (GDP) per capita (x-axis; unit: USD) and total publications on AI in the 1998–2017 period (y-axis; unit: publications); data from 18 countries across the economic spectrum were retrieved. Source: Elsevier [39].

**Figure 2 jcm-08-00168-f002:**
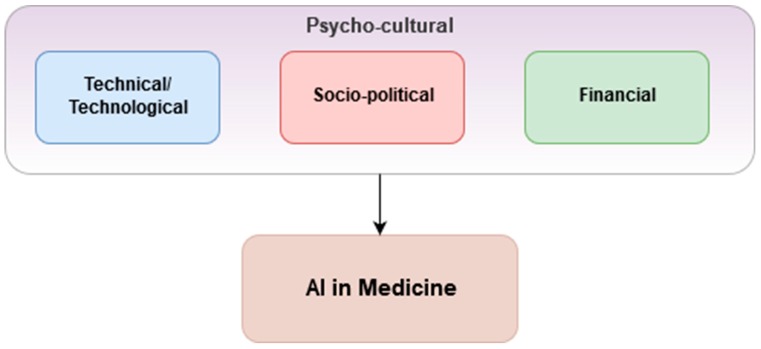
Four considerations for successful applications of artificial intelligence in medicine (AIM).

**Figure 3 jcm-08-00168-f003:**
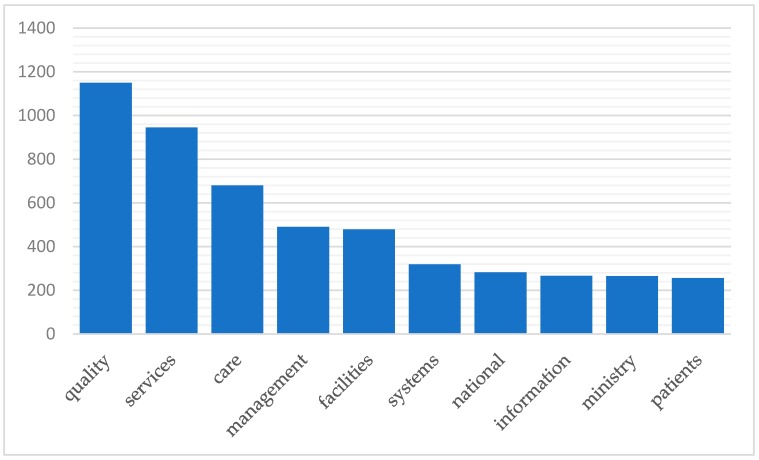
The top ten frequently used words (x-axis) in the Vietnamese Ministry of Health’s Joint Annual Health Report 2012 (y-axis: number of appearances). The results were obtained from the text mining of the ministry’s reports in R software (version 3.4.1).

**Figure 4 jcm-08-00168-f004:**
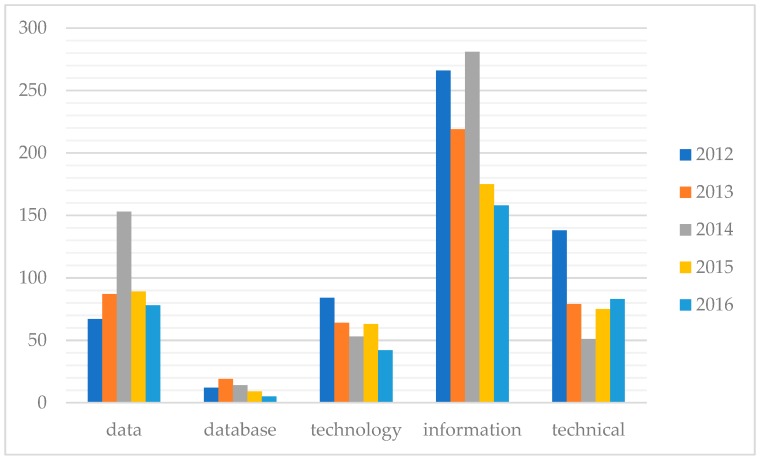
The frequencies of five keywords relevant to AI (x-axis) in Vietnam’s Ministry of Health’s annual reports from 2012–2016 (y-axis: number of appearances). The results were obtained from the text mining of the ministry’s reports in R software (version 3.4.1).

**Table 1 jcm-08-00168-t001:** Four major technological systems in AIM, as summarized by Ramesh, et al. [24].

	Artificial Neural Networks	Fuzzy Expert Systems	Evolutionary Computations	Hybrid Intelligent Systems
Inception date	1943: first artificial neuron;	1965: popularized by Lofti Zadeh;	1975: John Holland’s “Genetic Algorithms”;	
Description	Computational analytical tools;	Data handling methodology that permits ambiguity;	Computational techniques based on natural evolution process;	A combination of two or three of the above systems;
Algorithms	Multilayer feedforward; backpropagation algorithm; gradient descent;	Fuzzy control language; Continuous set membership from 0 to 1; “If-then” modeling: - Rule-based - Knowledge-based;	Stochastic search and optimization algorithms; Suitable for functions that are: non-linear, non-convex, non-differentiable, discontinuous;	Fuzzy logic; Genetic algorithms; Case-based reasoning; Neural networks;
Notable Applications	Clinical diagnosis; Image analysis in radiology & histopathology; Data interpretation in intensive care setting; Waveform analysis; Prognosis;	Diagnosis of certain types of cancer; Health/Clinical decision support systems; Reference consultation;	Outcome prediction in critically ill patients, lung cancer, melanoma, response to warfarin; Computerized analysis of certain carcinogenic diseases; Prediction of protein complexes;	Medical decision support tool; Breast cancer diagnosis; Diagnosis of coronary artery stenosis; Control of the depth of anesthesia;

**Table 2 jcm-08-00168-t002:** The probabilities of words that are associated with ‘database’ in the Vietnamese Ministry of Health’s Joint Annual Health Reports from 2012–2016. The results were obtained from the text mining of the ministry’s reports in R software (version 3.4.1).

	2012	2013	2014	2015	2016
0.32–0.38	lacking			information	analyzing
	standardization			inefficient	software
0.43–0.55			online	computers	
			warehouse	digital	
Above 0.56		inaccurate			
		standardizing			
		incomplete			
		inefficient			

**Table 3 jcm-08-00168-t003:** The utilization (or lack thereof) of information technology in Vietnam’s healthcare sector, categorized in three groups: administrative, clinical, and national.

Administrative	Progress	Clinical	Progress	National	Progress
Hospital management	54% of hospitals are using some software for internal management.	Diagnosis and treatment database	Under construction but remains not unified; hospitals use different software.	National standardized database	Not yet built.
Human resources management	Hospital software is said to be HR-focused.	Medical devices database	Under construction but not unified.	Departmental or provincial health databases	Under construction but not unified.
Database to track stakeholders in health management	Not yet built.	Drugs database/Prescription monitoring system	Under construction but largely mismatched.	Database about national target programs, preventive health, the private sector	Under construction but severely lacking.
Medical data collection	Manual collection, sporadic, lack of integration across establishments, which created duplication.	Electronic medical record (EMR) database	Under construction but needs to be standardized and integrated into the national system.	Health information system 2015–2020	Under construction but needs to be standardized.
Medical costs database	Not yet built.	Mortality and infection reports	Requested to be formulated a long time ago but lacking accurate data.	System to monitor implement of legal health documents	Not yet built.
Medical information reporting	There is no requirement for replacing paper reports with digital reports.	Database for diseases control	Not yet built.	Databases linking private and public health resources, and tracking patient responses	Not yet built.
Picture archiving and communication system (PACS)	Some medical establishments have used PACS but there is no official guideline for this yet.	Database to monitor adverse drug reaction (ADR)	Most hospitals have been collecting this information, but no official reports have been made.	Database about all licensed doctors	Not yet built.

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
