# Peer review of "Artificial Intelligence vs. Natural Stupidity: Evaluating AI Readiness for the Vietnamese Medical Information System"

_jcm, 2019, doi:10.3390/jcm8020168_

Reviewer 1 Report

This paper is very well written with a thorough evaluation of the research progress in the area of Artificial Intelligence and its applications, with a specific attention to the Healthcare sector. The manuscript covers quite a lot of ground, and presents an insightful overview beyond Vietnamese medical information system.  I do not really have any critical comments as everything, including presentation of figures, looks appropriate and has been carefully considered by the authors.

There is a misprint in line 100, where word ‘correction’ presumably should be ‘correlation’. Also, could authors check line 33 – the year of birth should be 1937 according to the Internet sources.

Author Response

Dear Sir/Madam,

We would like to take this opportunity to express our deep appreciation for providing very helpful and detailed comments for our manuscript. Your input has helped us improve our manuscript significantly. In this revision, we have addressed your concerns. Please notice that in the revised paper, the parts that are highlighted in yellow is for correction on the old text, the parts highlighted in green is written anew. Below are our answers to your comments (in italic). Also, the line numbers in the bold text refer to the revised paper.

Comments and Suggestions for Authors

This paper is very well written with a thorough evaluation of the research progress in the area of Artificial Intelligence and its applications, with a specific attention to the Healthcare sector. The manuscript covers quite a lot of ground, and presents an insightful overview beyond Vietnamese medical information system.  I do not really have any critical comments as everything, including presentation of figures, looks appropriate and has been carefully considered by the authors.

There is a misprint in line 100, where word ‘correction’ presumably should be ‘correlation’. Also, could authors check line 33 – the year of birth should be 1937 according to the Internet sources.

Thank you for your kind consideration of our paper. We have corrected for line 100 and line 33. In addition, we have also double-checked the English and made sure that no grammatical and wording mistakes are left. We have also made the following changes according to the requests of other reviewers:

We have added a brief discussion on the issue of “disease phenotyping and genotyping using big data analysis.”

“Within the virtual branch of AIM, specifically in bioinformatics, increasing attention is paid to the role of data-centric approaches, known as big data analytical tools and applications [27-33]. This focus is partly driven by the integration of genomic data into patient care, opening up the opportunities for clinical phenotyping patterns as well as posing serious challenges in the management of big data [28,31]. According to Chute, et al. [28], this development is “attributable to advances in chip-based genotyping and next-generation sequencing (NGS) methods.” The complex, large-scale data generated during clinical care, in addition to the electronic health records, can be useful in translational bioinformatics [32], cardiovascular disease research [27,31], deep patient genotype and phenotype identification [30], clinical pharmacology [33], or even in smartphone-based digital phenotyping to improve mental health [29]. In population-scale human phenotyping, the cautions remain that results in well-designed experiments are not always readily replicated and that there is increased burden of proof in mining data with layers of multiplicity [34].”

Regarding the update of the most recent studies, we have added three studies published in 2019.

Topol,      E. J. (2019). High-performance medicine: the convergence of human and      artificial intelligence. Nature Medicine, 25(1), 44-56.      doi:10.1038/s41591-018-0300-7

He,      J., Baxter, S. L., Xu, J., Xu, J., Zhou, X., & Zhang, K. (2019). The      practical implementation of artificial intelligence technologies in      medicine. Nature Medicine, 25(1), 30-36. doi:10.1038/s41591-018-0307-0

Faust,      K., Van Ommeren, R., Sheikh, A., Djuric, U., & Diamandis, P. (2019).      Deep learning for image analysis: Personalizing medicine closer to the      point of care AU - Xie, Quin. Critical Reviews in Clinical Laboratory      Sciences, 1-13. doi:10.1080/10408363.2018.1536111

In the Discussion and Conclusion, line 521-523:

“The challenge of data standardization is not unique to Vietnam as Faust, et al. [6] and He, et al. [9] have shown that this issue is among the critical barriers for the real-world clinical application of AI-technologies.”

Line 530-531:

“On the one hand, as investment into AI research or infrastructure for AI can be expensive, the irrational perspective of the public can slow down substantially the transitioning to an AI-powered healthcare sector [112], the experts community and science communication circle arguably must not ignore this problem of expectation management [105], as Topol [8] pointed out “The state of AI hype has far exceeded the state of AI science.”

Final Thank-you Notes

In closing this letter, we would like to extend our greatest gratitude to the hard work and time that you have put into improving our manuscript. Thanks to your valuable feedback, we were able to revise our manuscript to be more coherent and relevant.

We hope that the findings and insights in this study will be shared and serving the academic community as a whole. Please accept our sincere thanks for your great contributions to the advancement of sciences in the world.

Shall you have further questions, please do not hesitate to contact us. We look forward to hearing from you.

Best regards,

The authors

Reviewer 2 Report

   Overall this is a nicely presented and balanced review of the field. I suggest that authors should consider incorporating a number of recent publications to be up-to-date. Also, authors should consider inclusion of the “disease phenotyping and genotyping using big data analysis”, and a table summarizing recent publications (methods and results) could be very helpful to the casual reader. 

Author Response

Dear Sir/Madam,

We would like to take this opportunity to express our deep appreciation for providing very helpful and detailed comments for our manuscript. Your input has helped us improve our manuscript significantly. In this revision, we have addressed your concerns. Please notice that in the revised paper, the parts that are highlighted in yellow is for correction on the old text, the parts highlighted in green is written anew. Below are our answers to your comments (in italic). Also, the line numbers in the bold text refer to the revised paper.

Overall this is a nicely presented and balanced review of the field. I suggest that authors should consider incorporating a number of recent publications to be up-to-date. Also, authors should consider inclusion of the “disease phenotyping and genotyping using big data analysis”, and a table summarizing recent publications (methods and results) could be very helpful to the casual reader.

Thank you for your kind consideration of our paper. We have added a brief discussion on the issue of “disease phenotyping and genotyping using big data analysis.”

“Within the virtual branch of AIM, specifically in bioinformatics, increasing attention is paid to the role of data-centric approaches, known as big data analytical tools and applications [27-33]. This focus is partly driven by the integration of genomic data into patient care, opening up the opportunities for clinical phenotyping patterns as well as posing serious challenges in the management of big data [28,31]. According to Chute, et al. [28], this development is “attributable to advances in chip-based genotyping and next-generation sequencing (NGS) methods.” The complex, large-scale data generated during clinical care, in addition to the electronic health records, can be useful in translational bioinformatics [32], cardiovascular disease research [27,31], deep patient genotype and phenotype identification [30], clinical pharmacology [33], or even in smartphone-based digital phenotyping to improve mental health [29]. In population-scale human phenotyping, the cautions remain that results in well-designed experiments are not always readily replicated and that there is increased burden of proof in mining data with layers of multiplicity [34].”

Regarding the update of the most recent studies, we have added three studies published in 2019.

·        Topol, E. J. (2019). High-performance medicine: the convergence of human and artificial intelligence. Nature Medicine, 25(1), 44-56. doi:10.1038/s41591-018-0300-7

·        He, J., Baxter, S. L., Xu, J., Xu, J., Zhou, X., & Zhang, K. (2019). The practical implementation of artificial intelligence technologies in medicine. Nature Medicine, 25(1), 30-36. doi:10.1038/s41591-018-0307-0

·        Faust, K., Van Ommeren, R., Sheikh, A., Djuric, U., & Diamandis, P. (2019). Deep learning for image analysis: Personalizing medicine closer to the point of care AU - Xie, Quin. Critical Reviews in Clinical Laboratory Sciences, 1-13. doi:10.1080/10408363.2018.1536111

In the Discussion and Conclusion, line 521-523:

“The challenge of data standardization is not unique to Vietnam as Faust, et al. [6] and He, et al. [9] have shown that this issue is among the critical barriers for the real-world clinical application of AI-technologies.”

Line 530-531:

“On the one hand, as investment into AI research or infrastructure for AI can be expensive, the irrational perspective of the public can slow down substantially the transitioning to an AI-powered healthcare sector [112], the experts community and science communication circle arguably must not ignore this problem of expectation management [105], as Topol [8] pointed out “The state of AI hype has far exceeded the state of AI science.”

Regarding your request on a table summarizing the recent publications, we apologize for not being able to comply. We have referred to 70 papers on AI medicine published in 2014-2019. It is quite time-consuming to summarize their methods and results in a table. However, we have created an excel file titled “Review of Vietnamese Publications on AI” that give a summary of 47 AI-related papers on Vietnam. This file will be published as the Supplementary material.

Final Thank-you Notes

In closing this letter, we would like to extend our greatest gratitude to the hard work and time that you have put into improving our manuscript. Thanks to your valuable feedback, we were able to revise our manuscript to be more coherent and relevant.

We hope that the findings and insights in this study will be shared and serving the academic community as a whole. Please accept our sincere thanks for your great contributions to the advancement of sciences in the world.

Shall you have further questions, please do not hesitate to contact us. We look forward to hearing from you.

Best regards,

The authors

Reviewer 3 Report

I believe this paper is outstanding. It caught my attention, interest and curiosity.

It s really well written, and well organized.

Reading it was even so surprising, since I discovered so many interesting facts about the AIM research spread all over the word.

The authors not only clear explain their analysis and the findings of their analysis about the AI readiness for healthcare sector in developing countries, but also present a framework to perform this analysis in other countries. 

My only request is to add a table reproting all the abbreviations and their meaning.

Author Response

Dear Sir/Madam,

We would like to take this opportunity to express our deep appreciation for providing very helpful and detailed comments for our manuscript. Your input has helped us improve our manuscript significantly. In this revision, we have addressed your concerns. Please notice that in the revised paper, the parts that are highlighted in yellow is for correction on the old text, the parts highlighted in green is written anew. Below are our answers to your comments (in italic). Also, the line numbers in the bold text refer to the revised paper.

Reviewer’s comments:

I believe this paper is outstanding. It caught my attention, interest and curiosity. It’s really well written, and well organized. Reading it was even so surprising, since I discovered so many interesting facts about the AIM research spread all over the word. The authors not only clearly explain their analysis and the findings of their analysis about the AI readiness for healthcare sector in developing countries, but also present a framework to perform this analysis in other countries.

My only request is to add a table reporting all the abbreviations and their meaning.

Thank you for your kind consideration of our paper. According to your request, we have added a table reporting all abbreviations and their meaning in the Appendix:

Abbreviations

Short   for

ABC

artificial bee colony   optimization

ADB

Asian Development Bank

ADR

adverse drug reaction

AI

Artificial intelligence

AIM

Artificial intelligence in   medicine

ANNs

applications of artificial   neural networks

ARV

anti- retroviral

EMR

Electronic medical record

EWORS

Early Warning Outbreak Recognition   Systems

GIS

Geographic Information Systems

GLIF

GuideLine Interchange Format

IMCI

the Integrated Management of   Childhood Illness protocol

JAHR

Joint Annual Health Review

LSSVM

least squares support vector   machines

NAFOSTED

National Foundation for   Science and Technology Development

NB

Naïve Bayes models

PACS

Picture archiving and   communication system

PDAs

personal digital assistants

SVM

support vector machines

TCM

traditional Chinese medicine

Thank you very much for your consideration. We hope that the findings and insights in this study will be shared and serving the academic community as a whole. Please accept our sincere thanks for your great contributions to the advancement of sciences in the world.

Shall you have further questions, please do not hesitate to contact us. We look forward to hearing from you.

Best regards,

The authors

Round  2

Reviewer 2 Report

The authors answered carefully to our queries and revised their manuscript.  The manuscript has enhanced significantly, and I have no further comments.

Author Response

Dear Reviewer 2,
Thank you very much for providing very useful feedback for our paper. We greatly appreciate your time and consideration. We wish you the very best in your research.
Best regards, 

Ho Manh Tung